

# Identification of nighttime urban flood inundation extent using deep learning

Jiaquan Wan[1,2,3], Xing Wang[4], Yannian Cheng[5], Cuiyan Zhang[4], Yufang Shen[1,2,3], Fengchang Xue[5], Tao Yang[1,2,3] and Quan J. Wang[6]

[1]College of Hydrology and Water Resources, Hohai University, Nanjing, 210098, China

[2] The National Key Laboratory of Water Disaster Prevention, Hohai University, Nanjing, 210098, China

[3] Yangtze Institute for Conservation and Development, Hohai University, Jiangsu, 210098, China

[4] School of Computer Engineering, NanJing Institute of Technology, Nanjing 211167, China

[5] School of Remote Sensing and Surveying Engineering, Nanjing University of Information Science & Technology, Nanjing 210044, China

[6] Department of Infrastructure Engineering, Faculty of Engineering and Information Technology, The University of Melbourne, Victoria 3010, Australia

*Correspondence to*: Xing Wang (jwangxing0719@163.com)

**Abstract.** With the acceleration of urbanization, the disaster of urban flooding has had a serious impact on urban socio-economic activities and has become one of the important factors restricting social development in China. Accurate and timely identification of urban flooding extents is crucial for decision-making. Traditional remote sensing technologies are often limited by environmental factors, making them less suitable for application in complex urban terrains. The development of emerging technologies and the increase in urbanisation have led to a significant increase in the number of surveillance devices within cities, while the development of deep learning techniques has led to their widespread application in various fields. Deep learning methods using video images as a data source provide a new technical methods for intra-urban waterlogging recognition. However, current research mainly focuses on waterlogging recognition in daytime scenes, often ignoring nighttime, a time of high waterlogging incidence. To address these challenges faced by flooding recognition in the nighttime, this study proposes a deep learning model—NWseg—to achieve the recognition of the extent of waterlogging at night. Initially, we constructed a dataset of 4,000 images of nighttime urban flooding. Subsequently, MobileNetV2 and Resnet101 networks were used to replace the DeepLabv3+ backbone network and compared with the NWseg model. Next, the NWseg model is compared with ResNet50-FCN, LRASPP and U-Net models to evaluate the performance of different models in nighttime urban flooding identification. Finally, the applicability and performance differences of each model in specific environments were verified. In conclusion, this study successfully demonstrates the effectiveness of the NWseg model for nighttime urban flooding identification and provides new insights for nighttime urban flooding identification.

Keywords: Deep learning, Nighttime flooding identification, Urban flooding, NWseg



## 1 Introduction


In recent years, extreme rainfall events have been occurring frequently in the context of complex climate change.
Concurrently, with the acceleration of urbanization processes, the proportion of impervious surfaces has been continuously
expanding, resulting in serious urban flooding issues in many cities worldwide (Xue et al., 2023). Urban flooding often
coincides with multiple compounded disasters and may even trigger secondary calamities, posing serious threats to the safety
of urban residents, the normal operation of city functions, and sustainable development. This exacerbates the vulnerability of
urban socio-economic system (Luo et al., 2020). Therefore, achieving real-time and effective monitoring of urban flooding
has become a critical issue that urgently needs to be addressed.
Remote sensing technology has made significant advancements in the field of urban flood monitoring, providing new
perspectives for flood disasters identification through high spatial, temporal and spectral resolution data (Hao., 2022).
However, despite its excellent performance at the macro scale, remote sensing technology has limitations in urban area
monitoring. Due to insufficient temporal resolution as well as the influence of cloud cover and changing atmospheric
conditions, remote sensing techniques have difficulty in capturing subtle topographic changes within cities, and are unable to
monitor fast-changing flooding events in real time (Gao., 2023). In addition, the complexity of the urban environment,
especially the dynamic changes of small-scale water bodies and localized waterlogging, further increases the difficulty of
remote sensing technology in urban flood monitoring. Therefore, an intelligent and real-time urban flood monitoring method
is urgently needed to achieve more precise flood identification.
With technological advancements, the emerging fields of deep learning and computer vision have matured and engaged in
interdisciplinary collaborations, achieving remarkable results that offer new technical approaches for urban flood
identification. Particularly in image recognition, deep learning's advantages in extracting global features and contextual
information make it highly promising for inundation detection (Liao., 2023). Simultaneously, the increasing level of
urbanization has led to the widespread deployment of video surveillance devices across urban areas, particularly along city
roads, where they are ubiquitous. During rainfall, these cameras can fully record the flooding process, providing real-time
reflections of road inundation changes (Wang et al., 2024; Yang et al., 2022; Cheng et al., 2018). Therefore, combining deep
learning with traffic cameras can effectively achieve real-time recognition of urban flooding.
Existing research has demonstrated that deep learning excels in segmenting inundated areas. (Bai et al., 2021) utilized the
YOLOv2 object detection model to extract water accumulation features from images collected by Xi'an University of
Science and Technology, achieving an average recognition accuracy of over 85% through multiple model training sessions,
demonstrating the precision of this method for inundation area extraction. (Wang et al., 2021) classified road images into
four categories—background, dry surface, inundated area, and slippery surface—and used the Res-UNet+ semantic
segmentation network to handle different lighting and scene conditions, achieving an Mean Intersection over Union (MIoU)



of 90.07%, outperforming traditional classification methods. (Sarp et al., 2020) applied the Mask R-CNN model to
automatically detect and segment floodwaters in urban, suburban, and natural scenes, achieving 99% accuracy in the
detection phase and 93% in the segmentation phase. (Sazara et al., 2019) used a deep learning approach to detect standing
water on urban roads, in which a pre-trained VGG-16 network was used in the classification phase and a full convolutional
neural network was used in the segmentation task, and compared it with the traditional classifier and extraction algorithms
with manually-designed features, and the results showed that the deep learning approach has a more obvious advantage in
both the recognition and segmentation of standing water. However, current research focuses on daytime scenes, and the
existing datasets lack diversity to cover flooding scenes at night or under complex weather conditions. Meanwhile, some
algorithms underperform when processing images in low-light or adverse conditions, making flood identification at night or
in challenging weather a technical challenge. This limitation underscores the urgent need for accurate nighttime flood
monitoring and the necessity for algorithm improvements and dataset expansion.
For this specific scenario, we propose the NWseg model for waterlogging recognition in nighttime, inspired by the
method introduced by (Wei et al., 2023). The problem of insufficient model recognition accuracy in nighttime scenes is
effectively solved by two core components, Semantic-Oriented Disentanglement (SOD) and Illumination-Aware Parser
(IAParser) (Wei et al., 2023). On this basis, this study constructs an urban flooding dataset for nighttime scenarios, based on
which the model is trained to improve the model's ability to recognise the extent of urban flooding in nighttime
environments.
This study aims to enhance urban flood extent recognition in nighttime scenes by utilizing advanced semantic
segmentation techniques and a comprehensive all-nighttime dataset, addressing the current limitations in both datasets and
methodologies. More specific, our aims are as follows:
(1) Contributed a method for assessing urban flooded areas based on urban surveillance cameras in response to common
challenges in the field of nighttime urban flooding identification.
(2) A comprehensive and representative nighttime urban flooding dataset is constructed. It covers a wide range of
nighttime scenes, including different weather conditions and city layouts, providing a rich resource for training and testing
semantic segmentation models.
(3) Replacement of the original DeepLabv3+ model network backbone with MobileNetV2 and Resnet101 networks is
used to verify the performance impact of different network backbones on the DeeplavV3+ model through ablation
experiments.
(4) A waterlogging recognition model NWseg for nighttime scenarios is contributed, and the significant advantages of the
model in terms of robustness, effectiveness and practicality are verified by comparing with other existing models, which
advances the research and development of nighttime urban flood recognition.



## 2 Model

### 2.1 Nighttime Urban Segmentation Model

Nighttime scenes are typically characterized by low-temperature illumination and complex artificial light sources, which lead to changes in object appearance due to variations in lighting conditions. This reinforces the entanglement between light invariant reflectance and light-specific illumination, making it challenging to extract discriminative features for semantic segmentation. Based on this background, proposed a nighttime waterlogging recognition model —NWseg, specifically designed to cope with the problem of degraded segmentation performance due to insufficient illumination and complexity in nighttime scenes (Wei et al., 2023).

The paradigm consists of two core steps: decoupling and parsing. The inference is shown in Figure 1. In the decoupling phase, NWseg decomposes the input image into light-invariant reflectance and illumination-specific components. The designed SOD framework decomposes the image into illumination-independent reflectance components and light-specific components by semantically supervising the training of the de-entanglement module. It utilises Retinex theory to ensure that stable light-invariant reflectance is extracted under complex illumination conditions, which enhances the semantic recognition in the subsequent parsing phase. The parsing phase then extracts illumination features using an Illumination-Aware Parser (IAParser), which quantitatively evaluates the semantic information contained in the illumination by using a pyramid pooling module and a convolutional layer to construct an attention mask. The final segmentation result is obtained by combining reflectance and illumination features. The model effectively copes with the complex and variable lighting challenges in nighttime scenes through the dual mechanism of decoupling and parsing, and significantly improves the performance of semantic segmentation (Wei et al., 2023).

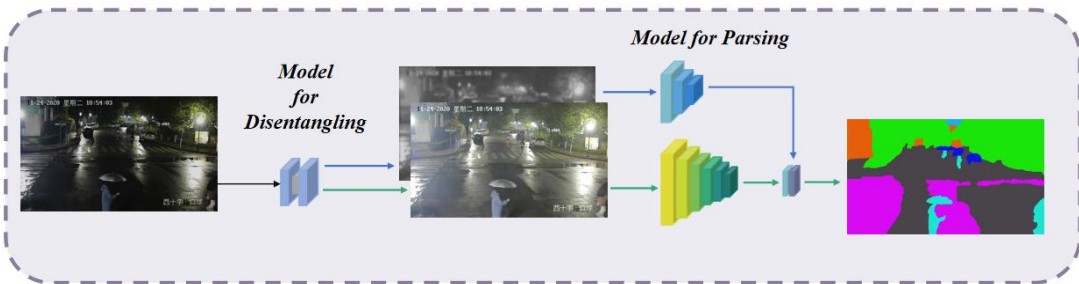

**Figure 1: NWseg model inference process**

### 2.2 Typical semantic segmentation model

The DeepLab network series is an improved set of models based on fully convolutional neural networks (FCNs). These methods effectively enhance the receptive field of convolutional kernels to acquire multi-scale feature information, thereby optimizing the spatial accuracy of segmentation results (Feng et al., 2023; Chen et al., 2024). The network models mainly utilize techniques such as atrous convolution and atrous spatial pyramid pooling (ASPP) to extract multi-scale features and



capture contextual information from images. The series includes DeepLabV1, DeepLabV2, DeepLabV3, and DeepLabV3+.
DeepLabV3+ is the latest version in the DeepLab series (Li et al., 2024; Peng et al., 2024; Ma et al., 2024; Zhang er al.,
2023); it introduces an encoder-decoder structure by adopting DeepLabV3 as the encoder and adding a decoder to form a
new model. The Xception model is applied to the segmentation task, extensively using depthwise separable convolutions
within the model. However, this network still has limitations in modeling long-range dependencies, insufficient handling of
class-imbalanced data, and higher latency for real-time applications. While DeepLabV3+ combines the spatial pyramid
pooling module and encoder-decoder structure in deep neural networks to achieve fine segmentation of object boundaries, it
remains constrained in modeling long-range dependencies, dealing with class imbalance, and reducing latency for real-time
applications (Li et al., 2023; Zhang et al., 2024; Tao et al., 2023).

To enhance the segmentation performance of DeepLabV3+ in urban flood scenes, this study designed ablation

experiments to verify the effectiveness of different backbone networks and compared them with the NWseg model. First,
experiments were conducted on the original, unmodified DeepLabV3+ network as a baseline model. Then, we replaced the
original DeepLabV3+ backbone network with the lightweight MobileNetV2, constructing an improved model (denoted as
MobileNetV2-DeepLabV3+). MobileNetV2 optimizes the number of model parameters by introducing a linear bottleneck
layer and inverted residual structures, ensuring a lightweight model while maintaining high accuracy (Jin et al., 2023).
Finally, we replaced the backbone network of DeepLabV3+ with the residual neural network ResNet101 to form another
improved model (denoted as ResNet101-DeepLabV3+). ResNet101 leverages a residual learning mechanism, allowing input
information to bypass certain layers, addressing gradient vanishing and explosion issues during deep network training. This
enhances the model's ability to capture spatial depth and details, ultimately improving the accuracy and robustness of flood
area recognition (Yang et al., 2023; Wang et al., 2024).

The Fully Convolutional Network (FCN) is an architecture specifically designed for semantic segmentation by replacing

the fully connected layers of traditional Convolutional Neural Networks (CNNs) with convolutional layers. This allows
FCNs to process input images of arbitrary sizes and perform accurate pixel-wise classification. FCNs extract features
through convolutional layers, reduce feature dimensionality via pooling layers, and restore feature map sizes using
upsampling layers, achieving precise pixel-level segmentation. Techniques such as bilinear interpolation are employed to
preserve image details ( Zhao et al., 2018). Additionally, skip connections in FCNs effectively fuse shallow and deep feature
information, improving segmentation accuracy. In this study, ResNet50 is adopted as the backbone network for FCN,
denoted as ResNet50-FCN. ResNet50 utilizes a residual learning mechanism to address gradient vanishing issues during
deep model training, maintaining training stability and efficiency while enabling greater depth. The multiple residual blocks
in ResNet50 capture rich multi-scale features, adapting to structures from coarse to fine. Its skip connections preserve the
detailed information that can be lost during upsampling, ensuring high-precision semantic segmentation. Combining the



depth of ResNet50 with the flexibility of FCN, this model enhances the accurate detection of inundated areas in complex
environments.
The LRASPP network is a lightweight semantic segmentation model designed for efficient operation on
resource-constrained devices such as mobile and embedded systems. It simplifies the classic ASPP (Atrous Spatial Pyramid
Pooling) module, retaining its ability to capture multi-scale contextual information while significantly reducing
computational complexity and memory usage. By leveraging depthwise separable convolutions to reduce the number of
parameters and incorporating detailed information from lower-level features, LRASPP achieves a balance between model
efficiency and accuracy. The model employs MobileNetV3 as the lightweight backbone to extract image features and
generate multi-scale feature maps. It also simplifies the original ASPP module by capturing multi-scale contextual
information through atrous convolutions and fusing low-level detailed features to improve segmentation accuracy. By
reducing convolutional layers and the number of channels, the network significantly lowers computational complexity. The
final output is upsampled to match the input image size, ensuring both efficiency and accuracy in segmentation tasks (Tang
et al., 2024).
U-Net is a classic network architecture for image segmentation, built on fully convolutional networks (FCNs). It utilizes
skip connections to directly concatenate features from downsampling and upsampling layers along the channel dimension,
effectively integrating information from different layers. U-Net features a symmetrical encoder-decoder structure, with a left
downsampling path, a right upsampling path, and intermediate skip connections. The downsampling path resembles
traditional CNN architectures, consisting of alternately stacked convolutional and pooling layers, while the upsampling path
uses transposed convolution to progressively restore the feature maps to the original image resolution (Zhang et al., 2023).
Shallow features primarily capture fine-grained information such as flood area edges, texture, and pixel position distribution,
while deeper features extract more abstract, coarse-grained semantic information, helping solve the final pixel-level
classification problem. U-Net's structural characteristics enable it to effectively handle detailed information in low-light
environments, making it particularly suitable for nighttime flood detection and other low-light image segmentation tasks
(Yadabendra et al., 2022).
We conducted comparative experiments on the FCN, LRASPP, U-Net, and NWseg models, evaluating their performance
using metrics such as Precision, Recall, Mean Intersection over Union (MIoU), and F1 Score. All models were initialized
with pretrained weights for their backbone networks and trained on the nighttime urban flooding dataset. The models were
then evaluated on the test set, with relevant metrics calculated to determine the most suitable model for nighttime urban
flood recognition.



## 3 Design of experiments

### 3.1 Construction of dataset

In this study, we employed web crawler technology using Google Chrome to construct a comprehensive nighttime urban waterlogging dataset by searching with the keyword "nighttime urban flooding." This dataset contains 4,000 images that capture a wide range of nighttime waterlogging scenes, varying in extent and shape. To enhance the dataset's robustness and comprehensiveness, we included images of complex scenes, such as strong lighting conditions and splashes caused by vehicles, ensuring its applicability to diverse nighttime flooding situations. During the data selection process, careful attention was given to the representativeness and balance of waterlogged areas across different scales, ranging from localized ponding to large-scale flood events, to ensure broad coverage of possible urban flooding conditions.

In addition, we performed the labeling work on the 4000 images in the dataset using the Labelme tool, which accurately extracted the waterlogged regions in each image. To further improve the accuracy of the annotations, we specifically assigned three graduate students to rigorously review and calibrate the boundary annotations for quality assurance. The annotation results are saved as labeled images. Figure 2 presents a comparison between the original images and the labeled images, where the waterlogged areas are marked in white and the non-waterlogged areas are marked in black.

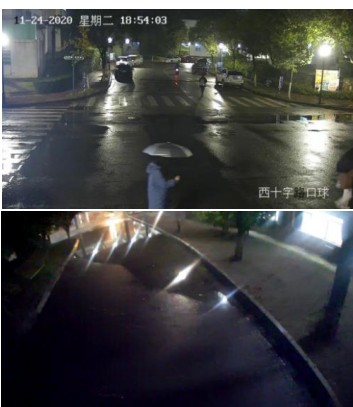
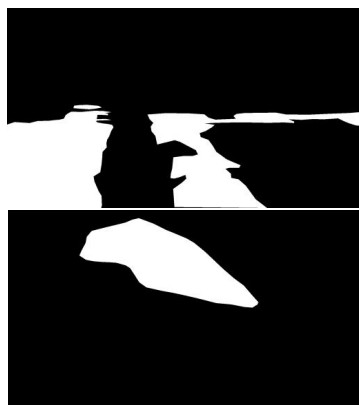

**Figure 2: Data Samples**

### 3.2 Evaluation metrics

In validation and testing, mean Intersection over Union (MIoU), F1score, precision and recall were used to assess the performance of the semantic segmentation models (Jin et al., 2024).

The MIoU value is defined as the ratio of the intersection area of the predicted bounding box and the real bounding box to the concatenation area, and is calculated by averaging the results for each category. It is used to evaluate the accuracy of the location information of the predicted results of the target detection task. The larger the overlap area between the real and the presumed area of the object, the larger the calculated value of MIoU, and the more accurate the presumed target area. The calculation of the MIoU value follows the following formula:



$$MIoU = \frac{1}{k+1}\sum_{i=0}^{k}\frac{\mathrm{TP}}{\mathrm{TP}+\mathrm{FP}+\mathrm{FN}}$$

(1)

Precision, which is the proportion of samples predicted to be positive that are actually positive, is also known as the check
rate, and can be expressed by the following formula:

$$Precision = \frac{TP}{TP+FP}$$

(2)

Recall, which is the proportion of actual positive samples that are predicted to be positive, is also known as the check all
rate, and is given by the following formula:

$$Recall = \frac{TP}{TP+FN}$$

(3)

F1score is the reconciled mean of precision and recall. The formula for each precision evaluation metric is as follows:

$$F1score = \frac{2\times Precison \times Recall}{Precison + Recall}$$

(4)

In the above formula, TP is the number of actual situations that are true and predicted to be true; FP is the number of
actual situations that are false and predicted to be true; FN is the number of actual situations that are true and predicted to be
false; and TN is the number of actual situations that are false and predicted to be false.
**3.3 Experimental configuration**
All    experiments    were    conducted    using    an    operating    system    of    Windows    10,    a    CPU    model    of
Intel(R)Core(TM)i712700F@2.10GHz, a GPU model of NVIDIAGeForceRTX3080, 32GB of operating memory,, a
programming language of Python 3.13, and a deep learning framework of PyTorch1.13, GPU acceleration libraries are
CUDA11.7, CUDNN8.4.1. the input image resolution is 512*512 pixels, the training optimizer type is Adam, the weight
decay index is 0.0001, and the initialized learning rate is 0.005. Parameters are shown in the Table 1.
**Table 1. Configuration table of the experiment**

| Project | Model |
| --- | --- |
| Operating System | Windows 10 |
| Programming Language | Python3.13 |
| GPU | NVIDIA GeForceRTX3080 |
| GPU memory | 32GB |



**4 Result**
**4.1 Ablation study**
**Table 2. NWseg and DeepLabv3+ series model training results**

| Models | P/% | R/% | F1score | MIoU/% |
| --- | --- | --- | --- | --- |
| Mobilenetv2-DeepLabv3+ | 67.46 | 50.64 | 57.85 | 46.15 |
| ResNet101-DeepLabv3+ | 67.74 | 57.24 | 62.05 | 51.98 |
| DeepLabv3+ | 53.34 | 50.61 | 51.94 | 46.07 |
| NWseg | 95.99 | 94.8 | 95.39 | 91.46 |

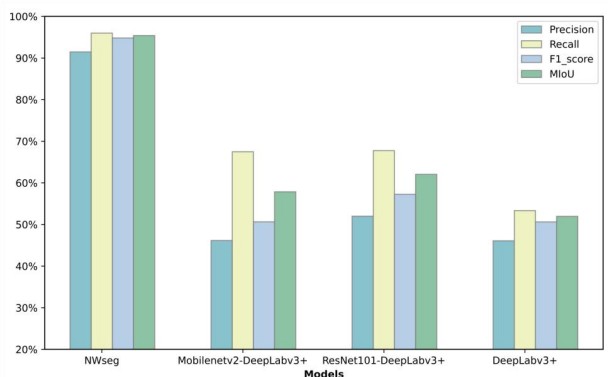


**Figure 3. Comparison of experimental results between NWseg and DeepLabv3+ series of models**
In this section, we present a comparative analysis of the DeepLabV3+ model with different backbone networks and compare
it with the NWseg model. As shown in Table 2 and Figure 3, replacing the original backbone of DeepLabV3+ with
MobileNetV2 resulted in improvements across all evaluation metrics. Precision and F1score increased significantly by
14.12% and 5.91%, respectively, while Recall and MIoU saw marginal improvements of 0.03% and 0.08%. When
ResNet101 was used as the backbone, the model's performance improved even more, with Precision, F1 score, Recall, and
MIoU increasing by 14.4%, 10.11%, 6.63%, and 5.91%, respectively. Despite these improvements, all three DeepLabV3+
models still exhibited a noticeable performance gap compared to the NWseg model. The NWseg model significantly
outperforms the other models by achieving 95.99%, 94.8%, 95.39%, and 91.46% in Precision, Recall, F1 score, and MIoU,
respectively.
Overall, the use of MobileNetV2 as the backbone network of DeepLabV3+ significantly improves the evaluation indexes
of the model while maintaining the lightweight, and MobileNetV2 successfully optimizes the computational efficiency and
reduces the consumption of computational resources, but its performance is far inferior to that of the NWseg model.The deep
network structure and advanced residual connection mechanism of ResNet101 make it perform more outstandingly in all
evaluation indexes. In contrast, ResNet101, with its deep network structure and advanced residual connection mechanism,
outperformed other backbones in all evaluation metrics, considerably boosting the overall performance of DeepLabV3+.
Nevertheless, even with ResNet101, the DeepLabV3+ models still lag behind the NWseg model, indicating there is
substantial room for further improvement in performance.





**4.2 Model performance experiments**
**Table 3.** NWseg and other segmentation model training results

| Models | P/% | R/% | F1score | MIoU/% |
|---|---|---|---|---|
| NWseg | 95.99 | 94.8 | 95.39 | 91.46 |
| ResNet50-FCN | 85 | 77.23 | 80.93 | 82.7 |
| Lraspp | 80.17 | 25.39 | 38.57 | 59.21 |
| U-Net | 94.7% | 83.57 | 88.24% | 80.5% |


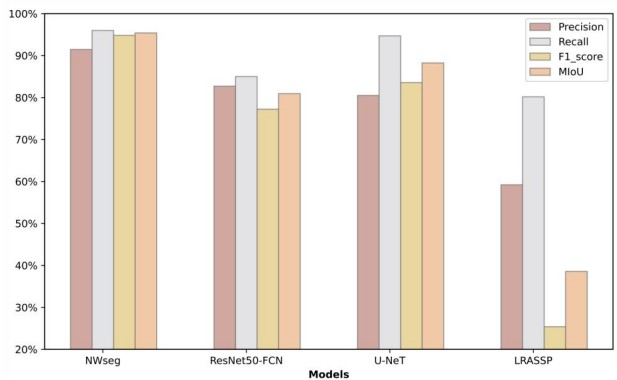


**Figure 4. Comparison of experimental results between NWseg and other segmentation models**
In this section, we present a comparative analysis of the experimental results of the NWseg model against other segmentation
models. As shown in Table 3 and Figure 4, the NWseg model achieved optimal results on the test set of the social inundation
dataset, with a Precision of 95.99%, Recall of 94.8%, F1-score of 95.39%, and MIoU of 91.46%. These metrics are
significantly higher than those of the other models, demonstrating exceptional accuracy and recall rates. Compared to the
ResNet50-FCN model, the NWseg model exhibits superior performance across all indicators, with increases of 10.99% in
Precision, 17.57% in Recall, 14.46% in F1-score, and an 8.76% improvement in MIoU. When compared with the U-Net
model, while the NWseg's Precision is similar, it outperforms in other metrics, with Recall, F1-score, and MIoU higher by
11.23%, 7.15%, and 10.96% respectively. Additionally, compared to the lightweight LRASPP model, the NWseg model
shows more pronounced advantages, with Precision increased by 15.82%, Recall significantly increased by 69.41%,
F1-score improved by 56.82%, and MIoU enhanced by 32.25%. The lightweight design of LRASPP limits its ability to
precisely capture details and edges, resulting in lower overall recognition accuracy.
Overall, the NWseg model demonstrates superior performance across all evaluation metrics and also shows strong
performance in real scenario tests. In contrast, although the ResNet50-FCN model performs well in precision and detail
processing, it lacks efficacy in handling edge regions, leading to slightly insufficient performance in complex scenes. While
LRASPP offers advantages in computational efficiency due to its lightweight design, it has limitations in the precise capture
of details and boundaries. The U-Net model is comparable to NWseg in accurately detecting target areas but is somewhat
less robust and consistent when processing complex scenes.



**4.3 Real-world scenes prediction comparison**
To validate the effectiveness and stability of each model under challenging scenes, we conducted tests on seven models
using nighttime rainfall scenes and nighttime strong illumination scenes (Liang et al., 2023). As shown in Figure 5(a)
presents the original scene where streetlights at night generate strong reflections and halos on the water surface. Additionally,
the intense lighting affects the detailed features of the ground. By comparing the recognition results of each model, it is
evident that the NWseg, ResNet50-FCN, and U-Net models accurately detected the flooding conditions in the scene. Notably,
the NWseg model exhibited a more refined recognition ability in identifying water accumulation in road depressions.
However, both ResNet50-FCN and U-Net showed certain false detections when recognizing the overall flooded areas. In
contrast, the Mobilenetv2-DeepLabv3+, DeepLab, and LRASPP models could only sporadically identify small flooded
regions and exhibited varying degrees of false detections. Although the ResNet101-DeepLabv3+ model recognized a larger
flooded area, a comparison with the original image reveals a relatively high false detection rate, indicating deviations in
prediction accuracy. Overall, the NWseg model outperformed the others in this scene recognition task, demonstrating
superior capability in recognizing flooded areas under complex lighting conditions.

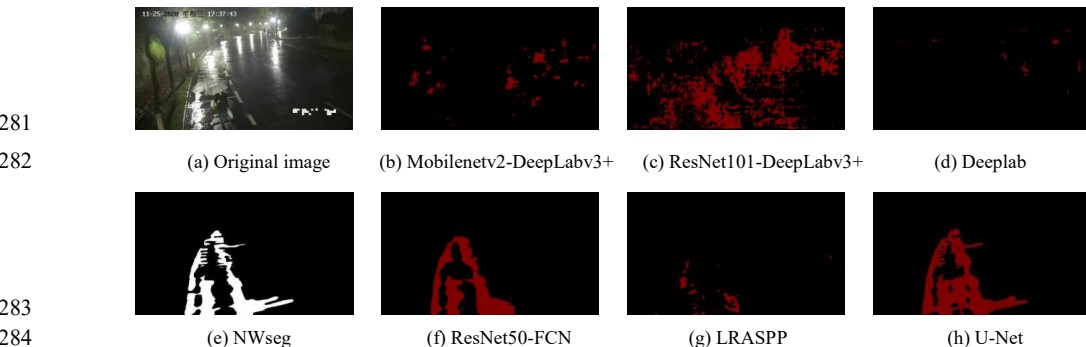


(a) Original image        (b) Mobilenetv2-DeepLabv3+        (c) ResNet101-DeepLabv3+        (d) Deeplab


(e) NWseg                    (f) ResNet50-FCN                    (g) LRASPP                        (h) U-Net

**Figure 5. Examination of nighttime strong illumination scenes**
Furthermore, in the nighttime rainfall scene tests, we evaluated each model's performance to simulate urban flood
recognition under real-world conditions (Tan et al., 2021). In such scenes, reflections from rainwater, slippery road surfaces,
and interference from raindrops on the camera lens can adversely affect image clarity and the models' recognition accuracy.
As shown clearly in Figure 6, the NWseg, ResNet50-FCN, and U-Net models were able to correctly identify the flooded
areas in the images, with the NWseg model providing the most detailed performance by accurately capturing the edges of the
flooded regions. While ResNet50-FCN and U-Net also identified the extent of flooding relatively well, they were somewhat
insufficient in recognizing the flood boundaries and exhibited some false detections.

In    contrast,    the    other    four    models    performed    relatively    poorly.    Specifically,    the    LRASPP    and

Mobilenetv2-DeepLabv3+ models were almost unable to detect the flooding, indicating weaker recognition capabilities in
nighttime rainfall scenes. Although ResNet101-DeepLabv3+ and DeepLab could detect some flooded areas, comparison



with the original images revealed that the regions identified did not accurately reflect the actual flooding conditions and had
high false detection rates. Through comparative analysis, we further confirmed the challenges posed by nighttime rainfall
environments for urban flood recognition and demonstrated the superior performance of the NWseg model in handling
complex conditions such as nighttime rainfall.

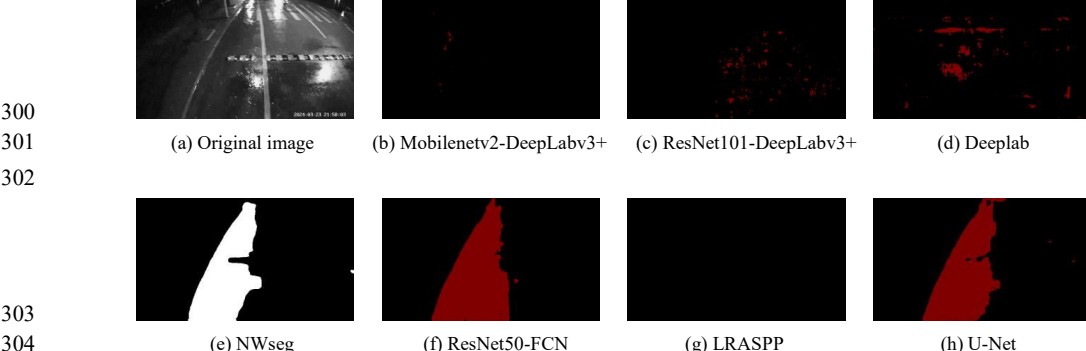

(a) Original image        (b) Mobilenetv2-DeepLabv3+      (c) ResNet101-DeepLabv3+              (d) Deeplab


(e) NWseg                   (f) ResNet50-FCN                   (g) LRASPP                      (h) U-Net

**Figure 6. Examination of nighttime rainfall scenes**
**5 Conclusions**
This study addresses the technical challenges of nighttime urban flood detection by evaluating the performance of seven
different models (Wan et al., 2024). First, we constructed a representative dataset comprising 4,000 images of nighttime
urban flooding scenes, covering various nighttime environments and diverse urban backgrounds. Second, a model for
nighttime waterlogging recognition, NWseg, is proposed to address the limitations in nighttime waterlogging recognition due
to insufficient lighting and complex lighting conditions. Furthermore, we replaced the backbone networks of the
DeepLabV3+ model with MobileNetV2 and ResNet101 and conducted ablation experiments to validate the performance of
DeepLabV3+ with different backbones in nighttime flood recognition. We also performed a comparative analysis between
these DeepLabV3+ models and the NWseg model, as well as systematically analyzed the NWseg, ResNet50-FCN, U-Net,
and LRASPP models. Based on this, we reached the following empirical findings:
(1) Within the DeepLab series, the DeepLabV3+ model using ResNet101 as the backbone outperformed other variants in
capturing water surface edges and shadow details.However, when compared to the NWseg model, there remains a
considerable performance gap.
(2) The NWseg, U-Net, and ResNet50-FCN models demonstrated excellent performance in recognizing large-scale
flooded areas, effectively capturing the overall contours of flood zones and exhibiting strong generalization capabilities.
Specifically, NWseg shows higher accuracy and robustness in complex scene tests, while ResNet50-FCN and U-Net have
some deficiencies and false detections in detecting edge details. In contrast, the lightweight LRASPP model showed limited
ability to recognize flooded areas in nighttime scenes, resulting in relatively poor performance.



(3) Through examining each model in complex scenes, we validated the NWseg model's effectiveness and stability in
diverse environments and conditions.
This study successfully demonstrates the superior performance of the NWseg model in nighttime urban flood
detection (Wan et al., 2024). However, the model's decoupling and parsing process involves complex decomposition of
lighting components and adaptive fusion, leading to high computational resource demands, which may impact its
practical usability. Future work will focus on reducing the model's parameters and computational costs while
maintaining accuracy. Additionally, further optimization of the dataset and model improvements will be pursued to
enhance the overall performance of the NWseg model, broadening its potential applications.
***Data availability.*** Data will be made available on request.
***Author contributions.*** **Xing Wang, Jiaquan Wan, Yannian Cheng, Cuiyan Zhang**: Writing – original draft, Validation,
Software, Methodology, Investigation. **Xing Wang, Jiaquan Wan, Yannian Cheng**: Writing – review & editing,
Validation. **Tao Yang**: Writing – review & editing, Supervision. **Fengchang Xue:** Formal analysis, Validation. **Yufang**
**Shen:** Data curation, Validation. **Quan J. Wang**: Data curation, Validation.
***Competing interests.*** The contact author has declared that none of the authors has any competing interests.

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
