# Peer review of "Identification of nighttime urban flood inundation extent using deep"

_EGUsphere, 2025_

## Author Comment (AC1)

Response to the Reviewers Comments

Author reply to RC1 egusphere-2025-77 (NHESS)

Paper title: Identification of nighttime urban flood inundation extent using deep learning

Authors: Jiaquan Wan et al.

**GENERAL COMMENTS**

This study proposes a method for recognizing flooded urban areas by analyzing night-time images taken by urban surveillance cameras during a rainy event. The method is based on combining deep learning model (using advanced semantic segmentation techniques) with video images as a data source to effectively perform real-time urban flood recognition, which addresses common challenges in the field of research into the identification of night-time urban flooding. The proposed Model - nighttime waterlogging recognition model —NWseg - outperforms the other models tested in this study for detecting urban nighttime flooding. The ability of the models to recognize flooding on a series of night-time flood images was used as a basis for validating the experimental results. My minor and major comments are presented in the next paragraphs.

**Response:** Sincerely thank the reviewer for the recognition and detailed evaluation of this study. We look forward to your feedback and commit to revising the manuscript earnestly based on your specific suggestions to further enhance its scientific rigor and readability. Below, the reviewer's comments are presented in black, and the authors' responses are provided in blue.

**Minor comments**

- Line 22: Please add space
- Line 33: It would be interesting to add references about complex climate change
- Line 219: Please remove comma repetition
- Line 240: Please add space
- Line 317: Please add space (Check along the text)

**Response:** We sincerely appreciate the reviewer's careful review and valuable comments. We will fully revise the manuscript according to the suggestions, including formatting corrections, removing redundant punctuation, and adding relevant references. Additionally, we will check the entire text to ensure consistency. Thank you for your detailed feedback, which has helped improve the quality of our manuscript.

**Major Comments**

1. Line 44 : "… remote sensing techniques have difficulty in capturing subtle topographic changes within cities …". Can the proposed model do it?

**Response:** We sincerely appreciate the reviewer's valuable comments and extend our sincere apologies for any potential misunderstanding caused by our wording. In the original manuscript (Line 44), we stated that "remote sensing techniques have difficulty in capturing subtle topographic changes within cities." However, this statement was not sufficiently precise and did not fully reflect the specific limitations of remote sensing technology in urban flood monitoring.

To directly address the reviewer's inquiry and clarify the capabilities of the NWseg model, We provide the following explanations. NWseg is not designed to directly capture subtle topographic changes within urban environments. Instead, it utilizes high-resolution video images obtained from mobile phones or urban surveillance cameras, to achieve real-time and fine-grained detection of urban flood areas. Unlike remote sensing technologies that rely on satellite data, NWseg is based on near-ground imagery, offering higher spatial resolution and temporal continuity. This enables the model to effectively capture the dynamic evolution of inundation areas, particularly in nighttime scenarios. By overcoming the limitations of remote sensing techniques—such as cloud cover interference and low spatiotemporal resolution—our approach provides timely monitoring support for rapidly developing flood events. In addition, Line 44 has been revised as follows: *"Due to the limitations in temporal resolution and the impact of cloud cover and atmospheric variations, remote sensing technology struggles to timely capture the dynamic changes of urban flooding, making real-time monitoring of rapidly evolving flood events challenging."*

2. Line 70 : "… under complex weather conditions …". Let's imagine that a heavy or exceptional nighttime rainstorm caused a power cut and consequently darkness in the city. What would be the performance (flood recognition) of the proposed NWseg model?
**Response:** We sincerely appreciate the reviewer's thoughtful question. For flood recognition under extreme nighttime conditions, such as a heavy rainstorm causing a power outage and complete darkness, the performance of the NWseg model largely depends on the availability of image data from surveillance cameras or mobile devices. While NWseg incorporates specialized preprocessing techniques and deep learning-based enhancements to handle low-light environments, its recognition capability would be significantly affected in the absence of any light sources.

In scenarios where minimal ambient lighting is available (e.g., vehicle headlights, emergency lighting, or infrared cameras), NWseg can enhance image quality through contrast enhancement and noise reduction techniques, thereby maintaining a certain level of detection performance. We acknowledge that in cases of complete darkness with no auxiliary lighting, the model's performance would be severely limited, which is an inherent constraint of image-based flood monitoring methods.

To address the reviewer's suggestion, we will introduce a new discussion section in the revised manuscript, where we elaborate on the model's robustness and potential future improvements. We sincerely appreciate the reviewer's insightful question, which has allowed us to further refine the applicability analysis of our approach and clarify future optimization directions.

3. Line 220 : "… GPU acceleration libraries…". What are the calculation times? Compatible with real-time forecasting? Please specify. Indeed the ability to provide results quickly to facilitate decision-making is essential in urban environments, given the economic, social, and other issues at stake.
**Response:** We sincerely appreciate the reviewer's detailed suggestion on Line 220. In the revised manuscript, we will include a comprehensive description of the GPU acceleration libraries, clarify the model parameters of NW-seg, and specify its response speed for flood

detection. We will also confirm its capability to support real-time prediction to meet the demand for rapid decision-making in urban environments.

4. Line 274 : An important element in the characterization of a flood is its depth, velocity and extent. It is not clear to me whether the model detects only the extent or both extent, velocity, and depth.
    - How does the NWseg model calculate depth? velocity?
    - To what spatial scale is the model applicable (Street? District? Whole city?)

**Response:** We sincerely thank the reviewer for their thorough examination and valuable suggestions regarding the content in line 274. In the revised manuscript, we will modify line 274 and related sections to clarify the model's capabilities and scope of applicability in response to the reviewer's specific comments. Specifically, the NW-seg model is designed to perform semantic segmentation on nighttime surveillance images, enabling precise identification of urban flood inundation extents. Furthermore, the spatial applicability of the NW-seg model depends on the coverage of the surveillance images. In this study, the model is based on street- and small-area-level surveillance imagery, making it suitable for street or community scales rather than city-wide applications.

5. Line 329 : "… Future work will focus on reducing the model's parameters…". The various model parameters have not been detailed.

**Response:** We express our heartfelt gratitude to the reviewer for their meticulous review and valuable feedback on the content in line 329. In the revised manuscript, we will include a detailed presentation of the parameters of each model in the Results section, accompanied by a comparative analysis. This will provide a foundation and direction for future efforts to reduce model parameters.

6. The study should robustly integrate a discussion section providing the advantages, limitations, and improvements of the proposed model.

**Response:** We sincerely appreciate the reviewer's valuable suggestions regarding the discussion section. In the revised manuscript, we will add a dedicated discussion section analyzing the advantages, limitations, and future improvements of the NW-seg model. First, this study constructed a dataset of 4,000 nighttime urban flood images, addressing the lack of nighttime scene data in existing research. Using this dataset, we conducted comparative experiments between NW-seg and other mainstream segmentation models, where NW-seg demonstrated superior performance across key evaluation metrics. To further validate its real-world applicability, we conducted tests in real-world scenarios. The experimental results indicate that, compared to other models, NW-seg exhibits significant advantages in detection stability and computational efficiency, demonstrating its adaptability and robustness in diverse nighttime environments.

However, despite NW-seg's strong performance in nighttime urban flood detection, certain limitations remain. First, while the dataset covers various nighttime flooding scenarios, the number of samples under extreme low-light conditions or strong dynamic reflections is relatively limited, which may affect the model's generalization capability. Additionally, in cases of total darkness due to power outages, NW-seg's detection performance is further

constrained. Second, due to the model's structural complexity, its computational resource requirements are relatively high, making it difficult to run efficiently on portable devices such as smartphones.

Future research will focus on optimizing the model to reduce parameter size and computational costs while maintaining detection accuracy, thereby improving its applicability in resource-limited environments. Furthermore, we plan to expand the model's deployment scenarios, such as integrating it with edge computing or lightweight network architectures, to enhance its generalizability and practical value in real-world applications.

---

## Author Comment (AC2)

Response to the Reviewers Comments

Author reply to RC2 egusphere-2025-77 (NHESS)

Paper title: Identification of nighttime urban flood inundation extent using deep learning

Authors: Jiaquan Wan et al.

The paper presents an interesting contribution on the use of deep learning to identify flood extents in urban areas from night-time images. As such it is relevant to NHESS, but it has a strong focus on the deep learning methods. Many readers of the journal, like myself, will not be familiar with the detailed language and concepts used in deep learning and the paper needs to be re-written with this audience in mind. Sections 2.1 and 2.2 need particular attention.

**Response:** We sincerely appreciate the reviewer's recognition of our study and the constructive feedback provided. In the revised manuscript, we will make comprehensive adjustments, particularly in Sections 2.1 and 2.2, by simplifying the language, reducing technical jargon, and providing additional background explanations to enhance readability for readers outside the deep learning field.

Much of the language in the paper is opaque and uses terms that are not common in scientific discourse (some of these are set out below). There also needs to be greater clarification of what the authors set out to do in the research, which of the methods they developed themselves, and what the conclusions mean for those working on urban flooding. For example, on Line 82 the aims are listed, but these are in fact a description of what was done without a justification. Further Line 74 says that NWseg is "proposed" and Line 91 says that the NWseg is "contributed", but was it developed by the authors or taken from other research work?

**Response:** We sincerely appreciate the reviewer's in-depth critique and constructive suggestions regarding the language and content of this paper. In the revised manuscript, we will comprehensively refine the language, clarify the research objectives and methodology, and further elaborate on the practical significance of our conclusions.

1. Clarification of Technical Terminology:

    To improve readability and adhere to scientific writing conventions, we will simplify complex deep learning terms such as illumination-invariant reflectance and disentanglement module. In the revised manuscript, we will provide clearer explanations in layman's terms to enhance accessibility for a broader audience.

*2. Refinement of Research Objectives:*

    We will clarify our research objectives in the introduction as follows:

    *This study aims to address the technical challenges of nighttime urban flood extent recognition and overcome the limitations of existing methods in low-light conditions. To achieve this, we first constructed a high-quality and representative nighttime urban flood dataset encompassing various lighting conditions and complex scenarios. Subsequently, we proposed a deep learning-based nighttime semantic segmentation model, NW-Seg, to enhance segmentation accuracy for flood-affected areas. Through comparative experiments with existing models, we systematically evaluated its effectiveness and applicability across different scenarios, providing a more reliable technological solution for nighttime urban flood monitoring.*

*3.* Clarification of Research Significance in the Conclusion:

We will refine the conclusion to better highlight our study's contributions as follows:

*Existing research primarily focuses on flood extent recognition in daytime scenarios, while nighttime flooding remains understudied due to the lack of datasets, complex lighting conditions, and the limited adaptability of current models to low-light environments. This study introduces a novel technological approach for the intelligent monitoring of nighttime urban flood extent. By addressing a critical research gap, our work not only advances the field of nighttime urban flood recognition but also serves as a reference for future deep learning applications in extreme lighting conditions.*

Through these revisions, we aim to enhance the clarity of our language, better define our research objectives, and highlight our contributions more explicitly, while ensuring a clearer impact on urban flood studies. Once again, we sincerely appreciate the reviewer's valuable feedback, which has significantly improved the quality and relevance of our manuscript.

Significantly more than half of the references that I tried to read online returned an error message or a message in Chinese characters.  Further, many of these are to non-peer reviewed sources.  Whilst the latter is acceptable in a few cases, the former is not at all acceptable in an international journal.

**Response:** We sincerely appreciate the reviewer's meticulous examination and constructive criticism regarding the references. Upon thoroughly reviewing the original reference list, we found that some links were invalid or directed to Chinese-language pages. This may be due to our citation of sources available only in Chinese academic databases or links that have become inactive over time. In the revised manuscript, we will remove all references that are inaccessible online or available only in Chinese and replace them with internationally recognized, peer-reviewed journal or conference papers accessible in English. Additionally, we have significantly reduced citations from non-peer-reviewed sources to ensure the accessibility and academic rigor of the references, aligning with the standards of international journals. We sincerely thank the reviewer for this valuable suggestion, which has significantly improved the academic quality of our paper.

**Specific points are:**

Line 50: a more scientific term that "remarkable" would be more appropriate.

**Response:** We will adjust the presentation of line 50 to ensure that the wording is more scientific and consistent with academic standards. In addition, we will check similar wording in the full text to ensure scientific and consistent language. We thank the reviewers for their valuable comments, and this suggestion significantly improves the academic rigor of the paper.

Line 54: it is stated that such surveillance is "ubiquitous", but whilst this may be true in the authors' experience is not true in all countries.  This should be acknowledged and it limits the usefulness of these methods.

**Response:** We sincerely appreciate the reviewer's careful attention to the wording in Line 54 and the constructive suggestion. In response, we have revised the corresponding expression in the manuscript and incorporated a discussion on this limitation.

In the original text at Line 54, we assumed the widespread presence of surveillance systems based on the research background, which primarily focuses on China's high-density urban monitoring networks. However, we overlooked the fact that in certain countries or regions, such as developing countries or low-density urban areas, surveillance coverage may be limited. To address the reviewer's concern, we have revised Line 54 to: *'Simultaneously, with the increasing level of urbanization, surveillance systems have been extensively deployed across most urban areas.'*

Additionally, we will expand the discussion section to analyze the current limitations of the model: The NWseg model relies on surveillance systems or mobile image data, which may limit its applicability in regions with weak monitoring infrastructure. Future research could explore the integration of drone imagery or other portable data sources to enhance its adaptability.

Line 55: of these three references two (Cheng, Yang) have links that do not work and one is to a publication that is not peer-reviewed. The remaining one does not mention whether this methodology has been tested in more than one country. Please clarify.

**Response:** We sincerely appreciate the reviewer's meticulous examination of the references. We acknowledge that some cited sources have accessibility issues, and some are not peer-reviewed. To address this, we will replace non-compliant references, prioritizing peer-reviewed journal articles and conference papers to enhance the reliability and academic rigor of the citations.

Line 89: the term "ablation" is common in machine learning, but is an example of a term that needs explaining to a different audience.

**Response:** We sincerely appreciate the reviewer's valuable feedback. We recognize that while 'ablation' is a common term in the field of machine learning, it may not be familiar to all readers of this journal. To enhance readability, we will provide the following explanation when the term is first introduced in the manuscript: *Ablation refers to the systematic removal or modification of specific components of a model to evaluate their impact on overall performance.*

Line 99: there is no subject in this sentence so we cannot see who proposed the model.

**Response:** We appreciate the reviewer's correction. We will revise the sentence to explicitly state that the model was proposed in this study, ensuring clarity and accurately attributing the research contribution.

Line 102: many terms in this paragraph need clarifying for a non-expert in machine learning, some examples are: SOD, illumination-independent reflectance, semantically supervising the training of the de-entanglement module, Retinex, Illumination-Aware Parser (IAParser), pyramid pooling module and a convolutional layer to construct an attention mask.

**Response:** We sincerely appreciate the reviewer's valuable feedback. We recognize that this paragraph contains multiple technical terms from the field of machine learning, which may pose challenges for readers unfamiliar with this domain. To address this, we will provide explanations for key terms in the revised manuscript, ensuring that the technical descriptions are simplified without compromising scientific rigor, thereby enhancing clarity and overall readability.

Line 115: a diagram showing how all these methods fit together would help readers understand what you are doing.

**Response:** We sincerely appreciate the reviewer's valuable suggestion. Line 115 of the manuscript primarily introduces other mainstream segmentation models to provide background for the subsequent performance comparison with our proposed NWseg model. The reviewer suggested using a diagram to illustrate the relationships among these methods for better comprehension.

We have carefully considered this suggestion; however, since this study focuses on the performance comparison between NWseg and other models rather than an integrated framework of different approaches, we believe that the textual description is already sufficiently clear and that an additional diagram is unnecessary. Furthermore, we will revise the abstract and introduction to explicitly clarify the comparison logic in the revised manuscript. We appreciate this suggestion, as refining these sections has significantly improved the logical clarity of the paper.

Line 189: what is a "Labelme tool"?

**Response:** We appreciate the reviewer's question. Labelme is an open-source image annotation tool widely used in image segmentation and object detection tasks. It enables users to manually annotate target regions in images by drawing polygons, thereby generating datasets for deep learning applications. In this study, Labelme was used to annotate the contours of flood-affected areas to construct a high-quality training dataset, enhancing the accuracy of the segmentation model. In the revised manuscript, we will further elaborate on this tool and its role to ensure better understanding for readers unfamiliar with it.

Line 191 says that the work was done by three graduate students.  Rather than describing who did the work, it is necessary to explain how they did it and how the quality of the analysis was checked and ensured.

**Response:** We sincerely appreciate the reviewer's detailed suggestions regarding Line 191. In the revised manuscript, we have removed references to who conducted the work and instead provided a detailed description of the workflow and quality control measures to enhance the scientific rigor and transparency of the study. In Section 3.1 Construction of Dataset, we have added the following clarification:

*We assigned three researchers with expertise in urban flood studies to independently annotate 4,000 nighttime flood images using the professional annotation tool Labelme, delineating the inundated areas. To further enhance annotation accuracy, all annotated images were cross-checked, and discrepancies were reviewed and adjusted to ensure*

*consistency. Finally, the annotated dataset was validated by experts in flood research to guarantee data quality.*

Line 193: "waterlogged" refers to soil saturation. I think "inundated" would be a better word.
**Response:** We sincerely appreciate the reviewer's detailed suggestions regarding the wording in Line 193. In the revised manuscript, we will correct any inaccuracies to ensure that the terminology aligns with the context of urban flood inundation.

Line 196: this and other figures captions need more details.
**Response:** We sincerely appreciate the reviewer's detailed suggestions regarding figure titles. In the revised manuscript, we will carefully refine the titles of all figures, incorporating additional details to enhance clarity and scientific precision.

Section 4.1: I think there should be less discussion of the three inferior methods as the differences between them are minor compared to their differences to NWseg.
**Response:** We sincerely appreciate the reviewer's suggestion regarding Section 4.1. In the revised manuscript, we will streamline this section by reducing the detailed comparison and analysis of suboptimal methods, instead emphasizing the overall superiority of NWseg and further highlighting its advantages.

Line 250: I don't see any experimental results in the text i.e. data that was collected through physical measurements on site.
**Response:** We sincerely appreciate the reviewer's thorough examination and valuable suggestions regarding Line 250. The original text states, 'Training results of NWseg, ResNet50-FCN, LRASPP, and U-Net models on the training set,' which describes the performance of these models on a night-time flood image dataset rather than data obtained through physical field measurements (e.g., water depth or flow rate).

    Additionally, the experimental data in this study are derived from an annotated surveillance image dataset rather than on-site physical measurements, aiming to assess the models' capability in identifying flood-affected areas. This aligns with the design objective of the NWseg model, which is to perform semantic segmentation based on surveillance imagery. In Section 4.3, Real-world scenes prediction comparison, we evaluated the models' performance in various complex scenarios. To further enhance the reliability of our assessments, we will manually annotate the flood extent in each scene and compare these annotations with the model predictions, providing a more comprehensive evaluation of model performance.

Line 252: what is "social inundation"?
**Response:** We sincerely appreciate the reviewer's careful examination and inquiry regarding the wording in Line 252. In our study, the intended term should be nighttime urban flooding dataset rather than 'social inundation.' In the revised manuscript, we will correct this terminology to ensure accuracy and scientific rigor while providing clearer context for the dataset.

Line 254: "exceptional" is too strong a word here.

**Response:** We sincerely appreciate the reviewer's detailed suggestions. In the revised manuscript, we will carefully review similar wording throughout the text to ensure scientific accuracy and consistency. We sincerely appreciate the reviewer's valuable suggestion, which has significantly enhanced the academic rigor and credibility of our paper.

Line 285: these images, and later one, are too small.

**Response:** We sincerely appreciate the reviewer's detailed suggestions regarding Line 285 and the related images. In the revised manuscript, we will adjust these images to enhance their clarity and readability, ensuring an effective presentation of the experimental results.

Line 328: if the conclusions state that there is a high computational demand, this should be investigated and reported on in the results section.   How much greater is it?   How long did it take?   What sort of computer was used?   Does this allow for practical use of NWseg?

**Response:** We sincerely appreciate the reviewer's thorough examination and valuable suggestions regarding Line 328. In response, we have added an analysis of NWseg's computational requirements in the results section and further clarified its practical implications in the conclusion to directly address the reviewer's concerns.

---

## Author Response (AR1)

**Response to the Reviewers Comments**

Author reply to RC1 egusphere-2025-77 (NHESS)

Paper title: Identification of nighttime urban flood inundation extent using deep learning

Authors: Jiaquan Wan et al.

**GENERAL COMMENTS**

This study proposes a method for recognizing flooded urban areas by analyzing night-time images taken by urban surveillance cameras during a rainy event. The method is based on combining deep learning model (using advanced semantic segmentation techniques) with video images as a data source to effectively perform real-time urban flood recognition, which addresses common challenges in the field of research into the identification of night-time urban flooding. The proposed Model - nighttime waterlogging recognition model —NWseg - outperforms the other models tested in this study for detecting urban nighttime flooding. The ability of the models to recognize flooding on a series of night-time flood images was used as a basis for validating the experimental results. My minor and major comments are presented in the next paragraphs.

**Response:** Sincerely thank the reviewer for the recognition and detailed evaluation of this study. We look forward to your feedback and commit to revising the manuscript earnestly based on your specific suggestions to further enhance its scientific rigor and readability. Below, the reviewer's comments are presented in black, and the authors' responses are provided in blue.

**Minor comments**

- Line 22: Please add space
- Line 33: It would be interesting to add references about complex climate change
- Line 219: Please remove comma repetition
- Line 240: Please add space
- Line 317: Please add space (Check along the text)

**Response:** We sincerely appreciate the reviewer's careful review and valuable comments. We have fully revised the manuscript according to the suggestions, including formatting corrections, removing redundant punctuation, and adding relevant references. Additionally, we checked the entire text to ensure consistency. In Line 31, we have added references about complex climate change as follows.

**References:**

Burn, D. H., Whitfield, P. H.: Climate related changes to flood regimes show an increasing rainfall influence, Journal of Hydrology, 617, 13, http://doi.org/10.1016/j.jhydrol.2023.129075, 2023.

Kim, H., Villarini, G., Wasko, C., Tramblay, Y.: Changes in the Climate System Dominate Inter-Annual Variability in Flooding Across the Globe, Geophysical Research Letters, 51(6), http://doi.org/10.1029/2023g1107480, 2024.

**Major Comments**

1. Line 44: "... remote sensing techniques have difficulty in capturing subtle topographic changes within cities ...". Can the proposed model do it?

**Response:** We sincerely appreciate the reviewer's valuable comments and extend our sincere apologies for any potential misunderstanding caused by our wording. We have improved this sentence as "Due to the limitations in temporal resolution and the impact of cloud cover and atmospheric variations, remote sensing technology struggles to capture the dynamic changes of urban flooding, making real-time monitoring of rapidly evolving flood events challenging."

2. Line 70: "... under complex weather conditions ...". Let's imagine that a heavy or exceptional nighttime rainstorm caused a power cut and consequently darkness in the city. What would be the performance (flood recognition) of the proposed NWseg model?

**Response:** Thanks to your suggestion, we have added the description about model performance in all-black scenarios in the new section "5 Discuss" as follows:

"On the other hand, in nighttime scenarios with extremely low illumination or even complete power outage (e.g., the case of city blackout triggered by heavy rainfall), the model has difficulty in extracting effective edge and texture information, which leads to a significant degradation of the recognition performance."

3. Line 220: "... GPU acceleration libraries...". What are the calculation times? Compatible with real-time forecasting? Please specify. Indeed the ability to provide results quickly to facilitate decision-making is essential in urban environments, given the economic, social, and other issues at stake.

**Response:** Thanks to your suggestion, we have added the description about the actual operating efficiency of the model in the new section "5 Discuss" as follows:

"In addition, NWseg achieves an inference speed of 37.8 FPS (i.e., approximately 26.5 milliseconds per image) under the NVIDIA GeForce RTX 3080 environment, demonstrating its potential for real-time applications in high-performance computing platforms."

- 4. Line 274: An important element in the characterization of a flood is its depth, velocity and extent. It is not clear to me whether the model detects only the extent or both extent, velocity, and depth.
  - How does the NWseg model calculate depth? velocity?
  - To what spatial scale is the model applicable (Street? District? Whole city?)

**Response:** Thanks to your suggestion, we have added the description about the applicable spatial scale of the model in the section "1 Introduction" as follows: "Meanwhile, Given that the data are mainly sourced from urban road surveillance systems, the method is particularly suitable for street (Street) and local area (District) scale flood detection." Furthermore, it is further clarified in the full text that this model is only used for flood extent identification tasks.

5. Line 329: "... Future work will focus on reducing the model's parameters...". The various model parameters have not been detailed.

**Response:** Thanks to your suggestion, we have added the parameter information of each model and provided a detailed analysis in the section "4 Result".

6. The study should robustly integrate a discussion section providing the advantages, limitations, and improvements of the proposed model.

**Response:** Thanks to your suggestion, we have added a new section "5 Discuss" as follows: "5 Discuss

In this study, a state-of-the-art model named NWseg is proposed to address the challenges of nighttime urban flood extent identification. Through a series of experimental validations, the NWseg model demonstrates superior performance with 95.99%, 94.8%, 95.39%, and 91.46% in Precision, Recall, F1 score, and MIoU, respectively. In the prediction comparison of real scenarios, the model also shows high accuracy and robustness, and effectively recognizes flooded areas in complex nighttime environments. In addition, NWseg achieves an inference speed of 37.8 FPS (i.e., approximately 26.5 milliseconds per image) under the NVIDIA GeForce RTX 3080 environment, demonstrating its potential for real-time applications in high-performance computing platforms. This study bridges the current research gap in flood extent recognition in nighttime scenarios, providing a technical reference for flood monitoring and emergency response.

Nevertheless, this study still has some limitations. First, the overall structure of NWseg is relatively complex, and the model parameters are large in scale, which limits its deployment capability on resource-constrained edge devices. On the other hand, in nighttime scenarios with extremely low illumination or even complete power outage (e.g., the case of city blackout triggered by heavy rainfall), the model has difficulty in extracting effective edge and texture information, which leads to a significant degradation of the recognition performance. In the future, we will further optimize the network structure to reduce the computational complexity of the model and improve deployment flexibility. In addition, we consider combining infrared thermal imaging, low-light image enhancement, or multimodal fusion methods to improve the robustness and generalization ability of the model under extreme low-light conditions."

The paper presents an interesting contribution on the use of deep learning to identify flood extents in urban areas from night-time images. As such it is relevant to NHESS, but it has a strong focus on the deep learning methods. Many readers of the journal, like myself, will not be familiar with the detailed language and concepts used in deep learning and the paper needs to be re-written with this audience in mind. Sections 2.1 and 2.2 need particular attention.

**Response:** We sincerely appreciate the reviewer's recognition of our study and the constructive feedback provided. We have revised Sections 2.1 and 2.2 to simplify the language, reduce the use of technical terms, and add relevant background information to enhance readability.

Much of the language in the paper is opaque and uses terms that are not common in scientific discourse (some of these are set out below). There also needs to be greater clarification of what the authors set out to do in the research, which of the methods they developed themselves, and what the conclusions mean for those working on urban flooding. For example, on Line 82 the aims are listed, but these are in fact a description of what was done without a justification. Further Line 74 says that NWseg is "proposed" and Line 91 says that the NWseg is "contributed", but was it developed by the authors or taken from other research work?

**Response:** We sincerely appreciate the reviewer's in-depth critique and constructive suggestions regarding the language and content of this paper. In the revised manuscript, we have comprehensively improved the language, clarified the research objectives and methods, and further elaborated on the practical significance of our conclusions. First, we simplified the complex deep learning terms in the paper. At the same time, it was made clear that NWseg was proposed by us. Secondly, we stated our research objectives in the introduction as follows:

- "(1) Contributed a method for nighttime urban flooding extent identification based on urban surveillance cameras, aiming at realizing efficient assessment of nighttime urban flooding areas and filling the gaps of research in this field at this stage.
- (2) To support the generalization ability of the model in complex nighttime environments, this study constructs a nighttime flood inundation dataset covering a variety of nighttime scenarios (e.g., different weather, illumination intensity, and urban structure), which provides diverse sample resources required for training and testing.
- (3) Replace the original DeepLabv3+ model network backbone with MobilenetV2 and ResNet101 networks and verify the effect of different network backbones on the performance of the Deeplavv3+ model.
- (4) An urban flood identification model NWseg for nighttime scenarios is proposed, and the significant advantages of the model in terms of robustness, effectiveness and practicality are verified by comparing with other existing models, which advances the research and development of nighttime urban flooding extent identification."

In addition, the significance of the research conclusions to urban flood researchers is clearly stated in the conclusion section as follows: "This study successfully demonstrates the superior performance of the NWseg model in nighttime urban flood detection, filling the research gap in nighttime flood range identification. Our work not only promotes the

development of the field of nighttime urban flood identification, but also provides a reference for future deep learning applications under extreme lighting conditions."

Significantly more than half of the references that I tried to read online returned an error message or a message in Chinese characters. Further, many of these are to non-peer reviewed sources. Whilst the latter is acceptable in a few cases, the former is not at all acceptable in an international journal.

**Response:** We sincerely appreciate the reviewer's meticulous examination and constructive criticism regarding the references. We have revised the references with invalid links or those pointing to Chinese pages, and added new references.

**Specific points are:**

Line 50: a more scientific term that "remarkable" would be more appropriate.

**Response:** Thanks to your suggestion, we have replaced "remarkable results" with "significant performance".

Line 54: it is stated that such surveillance is "ubiquitous", but whilst this may be true in the authors' experience is not true in all countries. This should be acknowledged and it limits the usefulness of these methods.

**Response:** Thanks to your suggestion, we adjusted the formulation of "ubiquitous" by revising it as follows: "*particularly in highly urbanized areas*".

Line 55: of these three references two (Cheng, Yang) have links that do not work and one is to a publication that is not peer-reviewed. The remaining one does not mention whether this methodology has been tested in more than one country. Please clarify.

**Response:** Thanks to your suggestion, we have added related references as follows.

**Reference:**

Hao, X., Lyu, H., Wang, Z., Fu, S., and Zhang, C.: Estimating the spatial-temporal distribution of urban street ponding levels from surveillance videos based on computer vision, Water Resources Management, 36(6), 1799-1812, https://doi.org/10.1007/s11269-022-03107-2, 2022.

Wang, Y., Shen, Y., Salahshour, B., Cetin, M., Iftekharuddin, K., Tahvildari, N., Huang, G.,
Harris, D., Ampofo, K., and Goodall, J.: Urban flood extent segmentation and evaluation
from real-world surveillance camera images using deep convolutional neural network,
Environmental Modelling & Software, 173, 105939,
http://doi.org/10.1016/j.envsoft.2023.105939, 2024.

Line 89: the term "ablation" is common in machine learning, but is an example of a term that needs explaining to a different audience.

**Response:** Thanks to your suggestion, we have added the description about ablation in the section "2.2 Typical semantic segmentation model" as follow: "this study designs a series of controlled experiments, systematically modifying or removing network components to verify the effectiveness of different backbone networks (i.e., ablation studies) and compared the results with the NWseg model."

Line 99: there is no subject in this sentence so we cannot see who proposed the model.

**Response:** Thanks to your suggestion, we have revised the manuscript to clearly state that the NWseg model is proposed in this study.

Line 102: many terms in this paragraph need clarifying for a non-expert in machine learning, some examples are: SOD, illumination-independent reflectance, semantically supervising the training of the de-entanglement module, Retinex, Illumination-Aware Parser (IAParser), pyramid pooling module and a convolutional layer to construct an attention mask.

**Response:** Thanks to your suggestion. In the section "2.1 Nighttime Urban Segmentation Model", we revised and clarified many deep learning terms.

Line 115: a diagram showing how all these methods fit together would help readers understand what you are doing.

Response: We sincerely appreciate the reviewer's valuable suggestion. Line 115 of the manuscript primarily introduces other mainstream segmentation models to provide background for the subsequent performance comparison with our proposed NWseg model. The reviewer suggested using a diagram to illustrate the relationships among these methods for better comprehension. We have carefully considered this suggestion; however, since this study focuses on the performance comparison between NWseg and other models rather than an integrated framework of different approaches, we believe that the textual description is already sufficiently clear and that an additional diagram is unnecessary.

Line 189: what is a "Labelme tool"?

Response: Thanks to your suggestion, we have added the description about Labelme in the section "3.1 Construction of dataset" as follows: "In addition, we employed Labelme, an open-source image annotation tool widely used in the field of computer vision, to manually annotate the flooded regions in the images. Through its graphical interface, annotators can polygonally map the inundated areas in an image and assign corresponding category labels to each area, thus generating high-quality semantic segmentation data that can be used for deep learning model training."

Line 191 says that the work was done by three graduate students. Rather than describing who did the work, it is necessary to explain how they did it and how the quality of the analysis was checked and ensured.

**Response:** Thanks to your suggestion, we have added the description about how the graduate students checked and ensured the quality of the annotations in the section "3.1 Construction

of dataset" as follow: "Specifically, each flood image was annotated separately by all three annotators, followed by a cross-review process to identify potential discrepancies in the flood boundaries. In cases of inconsistency, the annotators engaged in multiple rounds of collaborative discussion and iterative refinement, optimizing the boundaries based on image details."

Line 193: "waterlogged" refers to soil saturation. I think "inundated" would be a better word.

**Response:** Thanks to your suggestion, we have replaced "waterlogged" with "inundated".

Line 196: this and other figures captions need more details.

**Response:** We are very grateful for the reviewer's detailed suggestions on figure captions. We have improved all figures captions and added more details.

Section 4.1: I think there should be less discussion of the three inferior methods as the differences between them are minor compared to their differences to NWseg.

Response: Thanks to your suggestion, we have revised the section "4.1 Ablation Study" as follows: "In this section, we present a comparative analysis of the DeepLabV3+ model with different backbone networks and compare it with the NWseg model. As shown in Table 2 and Figure 3, all evaluation metrics are improved after replacing the original backbone network of DeepLabV3+ with MobileNetv2 and ResNet101, respectively. Notably, when ResNet101 was used as the backbone, the model achieved the best performance, with Precision, F1 score, Recall, and MIoU increasing by 14.4%, 10.11%, 6.63%, and 5.91%, respectively, compared to the baseline model. However, all DeepLabV3+ variants still exhibited a significant performance gap when compared to NWseg. The NWseg model achieved 95.99% in Precision, 94.80% in Recall, 95.39% in F1-score, and 91.46% in MIoU, demonstrating its superior capability in nighttime urban flood extent recognition. Although NWseg has a relatively large number of parameters, it delivers outstanding accuracy and robustness."

Line 250: I don't see any experimental results in the text i.e. data that was collected through physical measurements on site.

**Response:** We sincerely appreciate the reviewer's thorough examination and valuable suggestions regarding Line 250. The original text states, 'Training results of NWseg, ResNet50-FCN, LRASPP, and U-Net models on the training set,' which describes the performance of these models on a night-time flood image dataset rather than data obtained through physical field measurements. Additionally, the experimental data in this study are derived from an annotated surveillance image dataset rather than on-site physical measurements, aiming to assess the models' capability in identifying flood-affected areas. This aligns with the design objective of the NWseg model, which is to perform semantic segmentation based on surveillance imagery.

Line 252: what is "social inundation"?

**Response:** We sincerely appreciate the reviewer's careful examination and inquiry regarding the wording in Line 252. We have replaced "social inundation dataset" with "nighttime flood inundation dataset".

Line 254: "exceptional" is too strong a word here.

**Response:** Thanks to your suggestion, we have replaced "exceptional" with "superior".

Line 285: these images, and later one, are too small.

**Response:** Thanks to your suggestion, we have adjusted the sizes of Figures 5 and 6.

Line 328: if the conclusions state that there is a high computational demand, this should be investigated and reported on in the results section. How much greater is it? How long did it take? What sort of computer was used? Does this allow for practical use of NWseg?

**Response:** Thanks to your suggestion, we have added the description about the actual operating efficiency of the model in the new section "5 Discuss" as follows:

"In addition, NWseg achieves an inference speed of 37.8 FPS (i.e., approximately 26.5 milliseconds per image) under the NVIDIA GeForce RTX 3080 environment, demonstrating its potential for real-time applications in high-performance computing platforms."

---

## Author Response (AR2)

**Response to the Reviewers Comments**

Paper title: Identification of nighttime urban flood inundation extent using deep learning Authors: Jiaquan Wan et al.

We are grateful to the reviewers and the editorial staff for your valuable comments on this manuscript. We will respond to each of them and the relevant corrections are listed below. The relevant corrections have been highlighted in blue in the revised manuscript, and the responses to the detailed comments are provided below.

1. Expand the discussion on real-world applications and limitations.

**Response:** Thank you for your valuable suggestion. We have revised and expanded Section 5 "Discuss" in the the revised manuscript. The added content is as follows:

**Discussion on limitations:**

"In addition, the current model is primarily designed for nighttime flood extent recognition and is not yet capable of sensing or estimating flood depth. It also lacks the ability to perform reliably under all-weather conditions. Furthermore, although the NWseg model can identify flooded areas more accurately, it is still difficult to achieve accurate modeling and area quantification of inundated areas. Finally, the dataset used in this study is mainly collected from some typical cities in China, and although it has covered diverse nighttime environments and lighting conditions, the model's generalization ability may be limited by the influence of geographic concentration and the dependence on surveillance cameras, and there is a certain identification bias when facing areas with different urban structures and lighting conditions."

**Discussion on real-world applications:**

"In subsequent practical applications, the NWseg model can be widely deployed in key scenarios such as urban emergency management, intelligent transportation monitoring, and disaster prevention and mitigation, especially for emergency response needs under extreme weather at night."

2. Clarify how the model could be integrated into emergency response workflows or smart city infrastructure.

**Response:** Thank you for your valuable suggestion. We have clarified the potential integration of our model into emergency response systems and smart city infrastructure in Section 5 "Discuss" of the revised manuscript. The added content is as follows:

"By interfacing with existing traffic monitoring systems or urban sensing platforms, the model can automatically extract flooding information from the monitoring screen, and realize rapid identification and early warning push for waterlogged areas. Combined with the city scheduling platform, NWseg can assist government departments in flood risk assessment, dynamic allocation of emergency resources, and trend analysis of disaster evolution, which significantly improves the efficiency of urban response and risk management capabilities in extreme weather events."

3. Briefly comment on dataset generalizability and potential bias due to lighting or geographic scope.

**Response:** Thank you for your valuable suggestion. We have revised and expanded Section 5 "Discuss" in the the revised manuscript. The added content is as follows:

"the dataset used in this study is mainly collected from some typical cities in China, and although it has covered diverse nighttime environments and lighting conditions, the model's generalization ability may be limited by the influence of geographic concentration and the dependence on surveillance cameras, and there is a certain identification bias when facing areas with different urban structures and lighting conditions."

---

## Author Response (AR3)

Paper title: Identification of nighttime urban flood inundation extent using deep learning Authors: Jiaquan Wan et al.

Dear Editor,

I am writing to express my sincere gratitude to you and the reviewers for accepting our manuscript titled "Identification of nighttime urban flood inundation extent using deep learning" (Manuscript ID: egusphere-2025-77) for publication in Natural Hazards and Earth System Sciences. We greatly appreciate the constructive comments and valuable feedback provided during the review process, which have significantly contributed to improving the quality of our work.

Thank you once again for your constructive suggestions.

Sincerely!

Xing Wang and Co-authors.